# Health-Related Outcomes and Molecular Methods for the Characterization of A1 and A2 Cow’s Milk: Review and Update

**DOI:** 10.3390/vetsci11040172

**Published:** 2024-04-12

**Authors:** Alina Borş, Silviu-Ionuț Borş, Viorel-Cezar Floriștean

**Affiliations:** 1Department of Public Health, Faculty of Veterinary Medicine, “Ion Ionescu de la Brad” Iasi University of Life Sciences, 700489 Iaşi, Romania; alina.bors@iuls.ro (A.B.); viorel.floristean@iuls.ro (V.-C.F.); 2Research and Development Station for Cattle Breeding Dancu, 707252 Iaşi, Romania

**Keywords:** A1 milk, A2 milk, beta-casomorphin 7, bioactive peptides, dairy cows, human health

## Abstract

**Simple Summary:**

This review provides an update on the impact of A1 and A2 milk studies on milk consumption trends. Because it can cause worldwide panic and have very negative economic repercussions, the data from the updated literature must be interpreted extremely carefully. As this paper reveals, there are more in vitro and in vivo animal studies than human clinical trials. Clinical trials are more important because of the subject’s significance from the perspective of human health and because of the outcomes of in vitro and animal studies.

**Abstract:**

A new trend in cow’s milk has emerged in the market called type A1 and A2 milk. These products have piqued the interest of both consumers and researchers. Recent studies suggest that A2 milk may have potential health benefits beyond that of A1 milk, which is why researchers are investigating this product further. It is interesting to note that the A1 and A2 milk types have area-specific characteristics compared to breed-specific characteristics. Extensive research has focused on milk derivatives obtained from cow’s milk, primarily through in vitro and animal studies. However, few clinical studies have been conducted in humans, and the results have been unsatisfactory. New molecular techniques for identifying A1 and A2 milk may help researchers develop new studies that can clarify certain controversies surrounding A1 milk. It is essential to exercise extreme caution when interpreting the updated literature. It has the potential to spread panic worldwide and have negative economic implications. Therefore, this study aims to investigate the differences between A1 and A2 milk in various research areas and clarify some aspects regarding these two types of milk.

## 1. Introduction

The interest in differentiating between A1 and A2 beta-casein (β-CN) in milk dates to the early 1990s, subsequent to hypotheses positing that the β-CN variant in some bovine milk includes a peptide linked to an elevated risk of developing diseases such as type 1 diabetes mellitus (DM-1) and coronary heart disease (CHD) [1,2,3,4]. In a pivotal study published in 1992, Elliott reported a significant discrepancy in DM-1 incidence among Samoan children residing in New Zealand compared to their counterparts in Samoa, attributing this variance to differential milk consumption levels. A notable observation was the lower incidence of DM-1 among Maasai children in Kenya despite high milk consumption [3]. These observations underscored the imperative for rigorous investigation into the biochemical disparities between A1 and A2 β-CN and their potential ramifications on dietary recommendations and public health directives. The subject matter generated considerable interest beyond the academic community, attracting the attention of the media, the scientific community, and the business community alike. This widespread interest highlights the potential of these findings to impact nutritional science, public health strategies, and the dairy industry. Therefore, a multidisciplinary approach is necessary to further elucidate the implications of milk protein composition on human health.

Proteins, as substantial organic molecules, occupy a pivotal role in the architecture and functionality of all biological entities. Beyond their structural and mechanistic contributions, proteins are integral to the nutritional landscape of both animals and humans, serving as vital sources of energy, nitrogen, and indispensable amino acids [5]. Notably, dietary proteins may yield biologically active peptides, initially latent within their precursor sequences yet activated upon hydrolysis throughout food processing or digestion. Such bioactive peptides exert profound effects on human physiology, spanning from the modulation of gut motility and secretion to influencing blood pressure and encompassing antithrombotic, antioxidative, antimicrobial, and immunomodulatory actions. A subset of these peptides, engaging with the opioid receptor system, are, thus, categorized as opioid peptides. Sources of opioid peptides are diverse, including milk, cereals, vegetables, and meat products, with those derived from cow’s milk being the most rigorously examined [5,6]. Despite the predominance of in vitro and animal research, human clinical studies remain sparse and present incongruent findings, highlighting the necessity for more definitive scientific substantiation to elucidate the health implications of A1 versus A2 bovine milk [7]. This report aims to provide an updated synthesis of the critical dimensions of A1 and A2 cow’s milk. 

## 2. Milk Composition

Milk is defined as a whitish opalescent fluid containing milk proteins, fats, lactose and various vitamins and minerals, which is produced by the mammary glands of all adult female mammals after birth and serves as food for their young. Beyond its primary biological role, milk serves as the principal raw material in the production of diverse dairy products.

Cow’s milk contains approximately 32 g of protein per liter, with 80% casein protein (CN) and 20% whey protein (WP), based on their solubility at pH 4.6 [8,9,10]. The natural function of milk proteins is to provide young mammals with the essential amino acids needed for the development of muscles and other protein-containing tissues [11]. In dairy products, milk proteins also play a very important role, e.g., in processing, gelation, foaming, and other properties that may be considered desirable or undesirable in different applications [12,13]. Among milk proteins, the most important groups are CNs and WPs. The CNs are represented by αs1-CN, αs2-CN, β-CN, kappa-casein (κ-CN), and WP by the α-lactalbumin (α-Lg), β-lactoglobulin (β-Lg), immunoglobulins [14,15]. 

A free thiol group in β-Lg can be easily detected by thermal deconvolution. It is responsible for the formation of the disulfide bond that causes a heat-induced aggregation of WPs and between WPs and CN micelles in milk [16]. Whey proteins, especially β-Lg, can denature or aggregate and possibly associate with CN micelles via κ-CN bonds or form κ-CN/β-Lg complexes in the serum phase depending on pH as well as heat [17]. 

## 3. Casein 

The quality and safety of raw milk are influenced by its gross composition, number of somatic and microbial cells, and absence of various biological contaminants [18,19,20]. Recently, a single nucleotide polymorphism in the CSN2 gene, which is essential for β-CN coding, has been linked to raw milk quality based on genetics [21]. The four types of casein (αs1-, αs2-, β-, and κ-CN) are encoded by the genes CSN1S1, CSN1S2, CSN2, and CSN3, respectively [12,22]. They are mostly present in milk in the form of CN micelles, with the surface of the micelles consisting of the amphiphilic κ-CN, which stabilizes them against aggregation [23,24,25].

To date, the scientific literature has cataloged a spectrum of twelve β-CN genetic variants within bovine populations, specifically identifying A1, A2, A3, B, C, D, F, H1, H2, I, and G as distinct entities [26,27]. Predominantly, the variants A1 and A2 are observed, with B occurring less frequently, and A3, I, and C are categorized as rare. Notably, the E variant is exclusively identified within the Italian Piedmontese breed, signifying an important genetic marker. Moreover, the F variant has been identified within the Emilia Romagna region of Northern Italy, albeit at very low frequencies [27]. This genetic variability underscores the complexity of β-CN composition across different bovine breeds and geographies, presenting a rich area for further research in dairy science, particularly in relation to milk’s nutritional properties and its processing characteristics. Cows can express both types of genes in their milk, either homozygous or heterozygous with allelic co-dominance. This means that a cow with the homozygous A1/A1 genotype produces pure A1 milk, while the A2/A2 genotype produces pure A2 milk. However, if an individual is heterozygous for this gene, A1/A2, then the milk produced will contain a mix of the two forms β-CN. A1 and A2 milk is found in almost all *Bos taurus* populations [28]. The Holstein and Ayrshire breeds produce 63 and 67% A1 milk and 35 and 33% A2 milk, respectively [29].

The genetic features and breed of cows can influence the β-CN structure. According to the existing literature, the oldest genetic variant of β-CN is A2, meaning that initially, all herds had only one allele of this variety [30]. When comparing crossbred dairy cattle to the original native breeds, the first group has an A1 allele frequency that is higher than the other group [31]. However, the A2 allele is still the most common in many breeds, as examined in the extensive work by Sanchez et al. [32]. It was observed that crossbreed dairy cattle have a higher frequency of the β-CN A1 allele compared to their original native breeds. This observation suggests that genetic selection in crossbreed herds is based on protein yield, which is the total mass of protein obtained from a cow per day. It has also been noted that a higher protein yield is associated with the A1 variant [31]. Due to the new trend for A1-free milk, producers have started selecting females carrying the A2/A2 genotype. Currently, a variety of dairy products are being distinctly labeled to indicate the presence of the A2 β-CN protein, commonly referred to as A2 protein or A2 milk. This labeling practice is frequently associated with nutritional information, aiming to inform consumers about the specific type of protein contained within these products. Thus, bovine milk with β-CN A2/A2-carrying proline is A2/A2 milk (A2 β-CN genetic variant), while milk with histidine in β-CN A1/A1 or A1/A2 chains are A1/A1 or A1/A2 milk (A1 β-CN genetic variant, Figure 1) [33]. 

## 4. Distinction between A1 and A2 β-Casein

According to Hoque and Mondal [34], the A2 gene frequency is more common in many imported breeds of cattle, including Guernsey, Jersey, in the Channel Islands, and Asian and African cows. Conversely, the milk from Holstein Friesian cattle, which is the predominant dairy breed in regions such as Australia, Northern Europe, and the United States, primarily comprises A1 beta-casein. Notably, the Holstein breed exhibits a roughly equivalent distribution of both A1 and A2 β-CN variants. More than half of the Jersey breed exhibits the A2 β-CN variant, albeit with significant intra-herd variability. Similarly, over 90 percent of the Guernsey breed demonstrates the presence of the A2 β-CN variant. Since the early 1990s, scientific investigations have sought to elucidate the differences between the A1 and A2 β-CN variants, spurred by the discovery of a connection between the predominance of A1 β-CN proteins in milk and the occurrence of various chronic illnesses.

Upon comparative analysis, A2 beta-casein milk exhibits no visual differences from its A1 counterpart when evaluated in juxtaposition. To the novice observer, both variants present equivalent aesthetic and apparent quality characteristics. Despite their congruence in containing 209 amino acids, they diverge at the molecular level at the 67th amino acid residue: A1 milk is characterized by the presence of histidine, whereas A2 milk harbors proline at this critical juncture (Figure 2). This singular amino acid substitution is pivotal, influencing the health outcomes associated with the intake of A1/A1 versus A2/A2 β-CN milk. The underlying mechanism involves the differential enzymatic cleavage susceptibility at the 67th position; proline in A2 β-CN prevents cleavage during β-CN digestion, but histidine in the A1 variant permits proteolytic cleavage, releasing BCM-7 during β-CN digestion [35]. This distinction is critical in understanding the divergent health implications of consuming A1/A1 compared to A2/A2 β-CN milk, emphasizing the role of this amino acid in the post-digestive bioactive peptide profile [36]. Thus, BCM-7 is released exclusively from A1 and B variants, while A2 milk does not have the penchant for creating BCM-7 [37,38]. Proteases find it difficult to cleave the peptide bond within isoleucine and proline in the A2 milk between positions 66 and 67 due to increased enzymatic resistance; however, a different bioactive peptide, β-casomorphin-9 (BCM-9), results [39] (Figure 2). 

The European Food Safety Authority, in its 2009 evaluative report, identified BCM-7 as an opioid peptide specifically liberated from the digestion of A1 β-CN rather than A2 β-CN [5]. In the context of research into the digestive biochemistry of milk proteins, Asledottir et al. [42] and Cattaneo et al. [43] conducted simulated gastrointestinal digestion assays that led to the observation of BCM-7 release from A2 β-CN milk. Notably, the concentration of BCM-7 liberated from A2 milk was found to be significantly lower than that from A1 milk. Furthermore, emerging evidence suggests a potential biochemical interaction between BCM-7 and intestinal kappa-opioid receptors. Specifically, it was demonstrated that BCM-7, when administered in conjunction with enterostatin through a gastric cannula, can partially mitigate the reduction in fat absorption typically induced by a high-fat diet. This effect is attributed to the modulatory action of BCM-7 on kappa-opioid receptors, which may be an interesting receptor antagonist, as demonstrated by White et al. [44].

Recent investigations [45,46] highlighted the role of heat treatment in modulating the release of opioid peptides from bovine milk with the A1/A1, A1/A2, and A2/A2 β-CN genotypes during simulated gastrointestinal digestion processes. The research conducted by Daniloski et al. [47] emphasizes the disparities between the β-CN A1 and A2 phenotypes, extending beyond mere compositional differences to encompass divergent functionalities and environmental reactivities. Specifically, β-CN A2 is characterized by the formation of smaller micelles and the generation of less stable foams compared to β-CN A1. Technologically, milk containing β-CN A2 exhibits compromised acid gelation and rennet coagulation attributes, albeit demonstrating enhanced emulsion stability and foam formation capabilities. The findings underscore the critical necessity of identifying the variables influencing β-CN milk’s behavior throughout industrial processing, with significant implications for dairy processing and product formulation [47].

## 5. Health-Related Outcomes of A1 and A2 Milk

Recent research revealed that consuming A2 β-CN helped a low number of volunteers (*n* = 60) with their gastrointestinal issues and that regular milk drinking increased the amount of *Bifidobacterium* spp. in the distal colon. In association with this increase in the number of *Bifidobacterium* spp., improved symptoms of gastrointestinal discomfort were observed, such as a lower percentage of bloating and bowel movements, an increase in the frequency of bowel movements, and altered stool characteristics compared to uncharacterized regular milk [48,49]. In vitro studies demonstrated that BCM-7 stimulates the secretion of gastrointestinal mucin [50,51,52]. The oral administration of BCM-7 and consumption of A1 β-CN induced inflammatory responses in mice [53,54]. The A1 milk causes gastrointestinal inflammation, discomfort in the digestive tract, and/or delayed gastrointestinal transit in humans [55,56] and rats [57]. However, according to a study by Kappes et al. [58], feeding calves A1 or A2 milk did not affect their development, health parameters such as days with diarrhea or diarrhea occurrence, milk consumption, or body composition during the first two weeks of life. The process of β-CN digestion in the stomach, especially in the area that contains β-casomorphin peptides, was influenced by the β-CN genotypes. Nevertheless, following small intestinal transit, these variations disappear. Thus, it is necessary to investigate if genotype-dependent digestion has an impact on human digestion and overall well-being [59].

On the other hand, other studies found that BCM-7 improved intestinal mucosal immunity in mice [60,61] and rats [62] and that it had a positive effect on DM-1 using rats [63,64,65,66,67] and in vitro [67]. Also, Guantario et al. [68] show that mice’s intestinal immunity improved after consuming the A2 cow milk. The body of systematic reviews, including works by Daniloski et al. [21,47] and Kullenberg de Gaudry et al. [6,69], identified only limited evidence supporting the health advantages of A2 bovine milk in contrast to A1 milk. These findings were confirmed later by other studies [7]. The consumption of milk containing both types was associated with greater worsening of gastrointestinal symptoms and gastrointestinal transit time in lactose-intolerant humans than in lactose-tolerant humans, whereas milk containing only A2 β-CN did not exacerbate these symptoms in lactose-intolerant humans. These results suggest that the exacerbation of gastrointestinal symptoms associated with milk in lactose-intolerant humans may be related to A1 β-CN rather than lactose [70,71]. Further research is needed to determine the differential health effects between A1 and A2 milk types as the ongoing debates persist.

New Zealand, the Netherlands, Australia, the United Kingdom, and the United States are just some of the countries where commercially available A1-free cow’s milk is currently marketed as helpful for people with dairy intolerances. Today, commercial infant formula that is A1-free but contains CNs is widely sold in some regions such as China and Australia, with the claim that it is easier on the baby’s digestive tract [72]. However, A2 milk has been shown to be a safe substitute for human consumption, as no adverse effects of consuming A2 cow’s milk were demonstrated, and its nutrient content is comparable to that of A1 milk [73]. For this reason, A2 milk was first produced in New Zealand in 2003, and other countries such as the UK, Australia, the USA, the Netherlands, and China followed suit [27].

By 2029, the market for A2 milk in North America is expected to increase, a trend driven by consumer preference. This development reflects the general consumer trend towards a health-conscious diet, with A2 milk being seen as a better alternative due to its purported digestive properties [30]. Also, the European market will experience significant expansion as a result of the dairy industry’s efforts to conduct research and development. However, Oglobline et al. [74] reported that the Dietitians Association of Australia’s overall rationale was that all varieties of milk, A1/A1 milk, A1/A2 milk, and A2/A2 milk, have the same vital nutrients and are safe, nourishing, wholesome, and therefore, consumers are free to select the one that suits them.

Several studies found that a certain peptide is linked to an increased risk of various health issues, including DM-1, CHD, Sudden Infant Death Syndrome, and autism spectrum disorders [3,75]. It is important to note, however, that an association between the peptide and these conditions does not necessarily indicate causality. Additionally, some of the peer-reviewed literature suggests an inverse relationship between milk consumption and the risk of CHD and DM-1 [6,21,69,74]. Conversely, A2 milk is characterized by its beneficial impacts on human health, notably through the mitigation of gastrointestinal maladies [69,72]. Despite the increasing corpus of evidence, the linkage between A1 milk consumption and the etiology of neurological conditions, such as schizophrenia and autism, remains tenuously evidenced, with a prevailing hypothesis suggesting animal studies might provide an explanatory basis for these associations [6].

Investigations into the relationship between consumption of cow’s milk and the prevalence of DM-1 provided mixed results, with several studies concluding no interactive effect between early exposure to cow’s milk and the incidence of DM-1 [76,77,78]. This absence of a direct correlation might be attributed to the predominant role of enteral virus infections as a causative agent of DM-1 in pediatric populations [79]. However, research has shown that susceptibility to DM-1 varies with the protein content of cow’s milk, indicating that differences in milk protein profiles may contribute to DM-1 risk [80].

So, another important factor is the magnitude/amount of milk protein exposure in cows [81]. Though not yet established as the cause, there is an intriguing correlation between milk and DM-1, more specifically, A1 β-CN. Ecological evidence from various groups suggests that exclusive breastfeeding protects against DM-1 diabetes in young children. However, if the mother uses cow’s milk formula in addition to breast milk or breastfeeds for an excessively short period of time, this protection may be lost. Additionally, certain food triggers could enter breast milk. These variables may account for the discrepancies in the reported correlations between breastfeeding and DM-1. Vitamin D, for instance, might be a moderating or influencing element but not a causative one [79]. Milk protein’s diabetogenic properties were first demonstrated in BioBreeding (BB) rats, an animal model of spontaneous autoimmune diabetes. When rats were fed a typical laboratory meal, 50% of them acquired autoimmune diabetes (background rate), compared to 15% in rats fed a basic, semi-synthetic diet [2]. In a survey conducted between 1990 and 1994 in 19 developed countries, Laugesen and Elliot found a robust association between A1 β-CN consumption and the incidence of DM-1 [82]. Although many human feeding studies consistently reported the absence of BCM-7 in adult subjects, a limited number of investigations detected its presence in the bloodstream of infants [83]. While these observations do not establish BCM-7 as a causative agent for disease, they indicate a potential aggravation of health conditions in patients with pre-existing DM-1. However, the direct impact of A1 β-CN on DM-1 patients remains underexplored due to the absence of clinical trials specifically designed to assess its effects. 

A1 β-CN milk and CHD: McLachlan’s [4] investigation into the relationship between A1 β-CN consumption and mortality rates attributable to CHD revealed a significant correlation across 16 countries, highlighting a potential public health concern. In a related experimental investigation, Tailford et al. [84] demonstrated that rabbits fed with a diet containing 10% A1 β-CN for a duration of six weeks developed significantly larger areas of aortic fatty streaks compared to their counterparts fed with A2 β-CN, suggesting a differential impact of these β-CN variants on cardiovascular health. Laugesen and Elliott [82] further explored this connection, drawing from analyses from 20 developed countries. Their analysis, spanning two decades, revealed a positive and significant association between the per capita supply of A1 β-CN in cow milk and cream and the prevalence of ischemic heart disease. These findings collectively underscore the potential health implications of dietary A1 β-CN intake and its role in cardiovascular disease etiology. However, Chin-Dusting et al. [85] experimental outcomes did not reveal any significant variance in health markers between mice subjected to diets containing A1 versus A2 β-CN. Similarly, research undertaken by Venn et al. [86] assessed the impact of A1 and A2 dairy product consumption on human plasma cholesterol concentrations and found no significant differences in cholesterol levels between individuals consuming A1- and those consuming A2-type dairy products.

Other effects of β-CN include the following: Sun et al. [87] explored the possible link between the consumption of A1 milk and the occurrence of Sudden Infant Death Syndrome, which remains the leading cause of infant death. The study explored the hypothesis that β-casomorphins, peptides resulting from the digestion of A1 β-CN, might pass into the infant’s central nervous system. They hypothesized that these compounds might exert an inhibitory effect on the respiratory center located in the brainstem, potentially leading to apnea and subsequent death. Exogenous peptides can inhibit peptidease/enzyme activity, thereby increasing the activity of endogenous opioids. Thus, the excessive absorption of bioactive peptides such as bovine or human β-casomorphins may be of pathological significance in some infants at risk of Sudden Infant Death Syndrome. However, to establish this hypothesis, it is necessary to measure both endogenous and exogenous opioid peptides in the plasma and cerebrospinal fluid of neonates with respiratory disturbances and then compare these concentrations with those in healthy individuals [87].

Lucarelli et al. [88] suggested that cow’s milk consumption could exacerbate behavioral symptoms in children with autism, a hypothesis further sustained by the works of Reichelt and Knivsberg [89], who detected opioid peptides originating from food proteins in the urine of autistic patients. In a bibliometrics and text mining study initiated by Gonzales-Malca et al. [7], the authors of a studied cluster found a link between BCM-7 and autism in children [90]. Additionally, they discovered that BCM-7 had a positive effect on the behavior and learning of rats, as evidenced by previous studies [91,92,93,94]. This connection was confirmed through the detection of bovine BCM-7 in autistic children’s urine, employing a highly sensitive ELISA technique [88]. Conversely, studies by Hunter et al. [95] and Cass et al. [96] reported no detection of such peptides in autistic children, presenting a contrasting view. Additionally, later studies evaluate the role of BCM-7 as an immunomodulatory agent and its correlation with autism and atopic dermatitis, highlighting the complex interplay between dietary proteins and certain health conditions [97,98,99].

Further research demonstrated that the β-CN variants A1, A2, and B produce distinct peptide profiles upon in vitro digestion. This phenomenon was explored by Lisson et al. in their 2013 publication [100], where they hypothesized that the resultant peptide variants from each β-CN type could exhibit unique affinities for immunoglobulin E, influencing the binding process. Such differential binding activities suggest a nuanced immunological interaction that may influence allergenicity among the β-CN variants. Other investigations revealed an increase in the antioxidant properties of β-CN post-digestion [101,102,103]. These findings highlight the significant impact of β-CN digestion on enhancing its antioxidative functionality, offering insights into the protein’s contribution to health-promoting dietary components [104]. 

Although animal studies frequently reported favorable health outcomes associated with A2 milk consumption, these findings have not been consistently validated by human clinical trials, as noted by Gonzales-Malca et al. [7]. There is, however, a widespread consensus regarding A2 milk’s ability to mitigate digestive intolerance symptoms commonly linked to A1 milk consumption. Despite this consensus, Brooke-Taylor et al. [72], Gonzales-Malca et al. [7], and Jeong et al. [105] advocate for further clinical research to explore the effects of A1 milk consumption. Such studies should encompass a broad spectrum of demographic variables, including age, ethnicity, and genetic predispositions, as well as various nutritional statuses, to thoroughly understand the impacts of A1 milk across heterogeneous populations.

## 6. New Molecular and Biochemical Methods for Characterization A1/A2 Milk

Characterizing bovine herds using CSN2 genotyping tests has garnered a lot of attention in recent years. This is supported by the growing demand for only A2 dairy products, which necessitates the use of cows with A2/A2 genotypes exclusively. In this regard, rhAmp^®^ SNP genotyping and high-resolution melting (HRM) were evaluated by Giglioti et al. [106] as two methods to identify the CSN2 gene alleles in milk samples. These authors claim that genotypes from the CSN2 gene in milk can be distinguished using either approach. Nevertheless, rhAmp was ten times more sensitive than HRM at identifying the presence of A1 milk in a sample that contained only A2. The same research group developed another accurate and specific method based on real-time PCR for directly detecting A1 and A2 alleles in milk [107]. Two years later, Watanabe et al. [108] developed a highly sensitive PCR-based method capable of detecting the A1 allele in A2 milk samples without necessitating DNA isolation. The method leverages the CycleavePCR technique to directly amplify the β-casein (β-CN) gene from raw milk samples. The genotypic analysis of milk samples produced results that were in complete agreement with those obtained from genomic DNA, demonstrating the method’s reliability. Furthermore, this approach successfully identified the presence of the A1 allele in A2 milk samples with a detection threshold of 2%, indicating its good sensitivity and offering a highly precise tool for dairy genetics research.

Liu et al. [109] developed an analytical technique using liquid chromatography–mass spectrometry (LC-MS) to differentiate A1 and A2 β-CN in milk. The technique targets characteristic thermolytic peptides and uses an optimized chromatography gradient to fully separate the A1 and A2 peptides. The efficacy of the technique was demonstrated through the analysis of four traditional and two A2 milk samples. The study confirmed that the A1 β-CN variant is dominant in regular milk samples. One of the two A2-labeled samples did not exhibit any A1 β-CN variant, while a trace amount was identified in the other. This highlights the high sensitivity, specificity, and operational efficiency of the novel method.

The potential for high-quality DNA yields in large-scale screening led to the evaluation of both manual and automated DNA extraction techniques, which were evaluated in a recent work by Vigolo et al. [110]. The objective was to evaluate the effectiveness of such techniques in differentiating the genetic variants A1 and A2 of β-CN starting from milk somatic cells, as well as the operational cost and time required for analysis, expertise, and labor requirements. First, the most effective way to obtain large amounts of total genomic material has been shown to be automated DNA extraction from a complex matrix, such as milk. Second, the most effective and high-quality method for indirectly genotyping the cow was demonstrated by the high-performance liquid chromatography (HPLC) approach. It does, however, necessitate more analysis time. As a result, polymerase chain reaction–restriction fragment length polymorphism (PCR-RFLP), and amplification refractory mutation system–polymerase chain (ARMS-PCR), were shown to be an extremely dependable technique for characterizing the most prevalent β-CN variants (A1/A2). The analysis took less time with the molecular methods. Specifically, the ARMS-PCR proved to be the most affordable, rapid, and intuitive approach. Finally, a more involved and expensive process was needed to ascertain the cow’s genotype using the PCR-RFLP method, which depends on the use of a restriction enzyme [110]. Although more expensive than HPLC analysis, both allele-specific PCR techniques proved to be quick and accurate in differentiating between A1 and A2 variants. To be more precise, out of all the techniques that were evaluated, PCR-RFLP was the most costly and labor-intensive, while ARMS-PCR was the fastest and required less technical expertise. All things considered, ARMS-PCR, in conjunction with automated milk matrix DNA extraction, is the most appropriate method for large-scale genetic CSN2 gene characterization Vigolo et al. [110].

Dairy processing conditions can trigger reactions between proteins and reducing sugars, known as the Maillard reaction. This reaction produces a multitude of proteoforms characterized by the attachment of one or more sugar moieties [111]. In a further exploration of protein composition’s role in dairy processing, Danilosky et al. [112] demonstrated that the genetic variant of β-casein significantly affects the physical properties and digestive behavior of skim milk powders. These studies reveal the significant impact of manufacturing processes and protein genetics on the functional and nutritional characteristics of dairy products. The ultraperformance liquid chromatography–high-resolution mass spectrometry (UPLC-HRMS) intact protein method presented in recent studies [113,114,115] is an effective method for the identification and quantification of proteins in dairy samples such as raw milk, skim milk powder, whey powder, final products, and samples from other species such as buffalo or human. If a protein’s overall signal is adjusted for a factor associated with its glycation index, it is feasible to quantify individual proteins in processed food matrices [113]. The authors, therefore, noted that the major milk proteins (below 30 kDa) and their corresponding proteoforms can be easily and unambiguously detected using the UPLC-HRMS method.

## 7. Conclusions 

The emergence of comprehensive genotyping and genomic selection technologies revolutionized our ability to analyze the frequencies of milk protein variants on an unprecedented scale. While academic literature reviews serve as an indispensable resource in this field, their inherent subjectivity—stemming from the selection bias of the researcher towards certain studies—may overlook significant contributions. Promoting A2 milk consumption, while driven by potential health benefits, necessitates a balanced approach considering the implications for genetic diversity among dairy cattle. The sustainability of dairy farming, conservation of genetic resources, economic viability, ecosystem health, and future adaptability all hinge on maintaining a broad genetic base. Therefore, strategies that promote A2 milk should also incorporate measures to preserve the genetic diversity of A1 milk breeds, ensuring a holistic approach to dairy production that safeguards both public health interests and biodiversity. This analysis shows a discrepancy in the body of research, with a greater focus on in vitro and in vivo studies involving animals rather than human clinical trials. Considering the paramount importance of human health and the limitations inherent to in vitro and animal models, the need for extensive human clinical trials becomes evident, highlighting the critical nature of translating these preliminary findings into concrete health outcomes for the human population.

## Figures and Tables

**Figure 1 vetsci-11-00172-f001:**
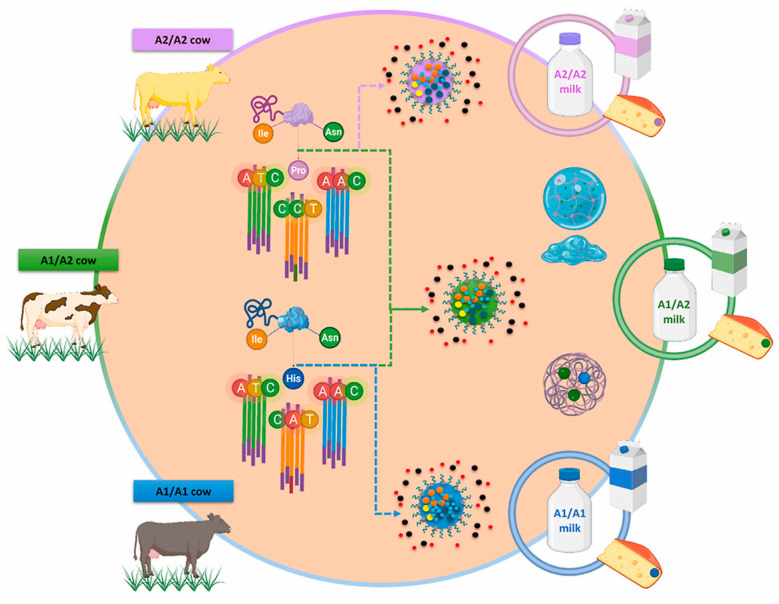
Bovine genetic polymorphism; variation sequences of A2 β-casein genetic variant and A1 β-casein genetic variant. Ile (Isoleucine), Pro (Proline), His (Histidine), Asn (Asparagine); A (Adenine), T (Thymine), C (Cytosine) [33].

**Figure 2 vetsci-11-00172-f002:**
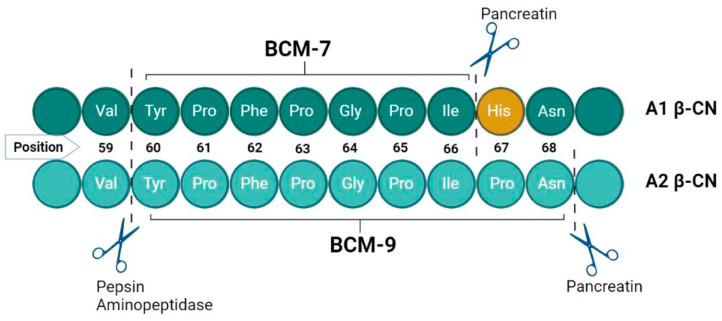
Proteolytic digestion of A1 β-casein and A2 β-casein in terms of BCMs release [40,41].

## Data Availability

No new data were created or analyzed during the development of this manuscript.

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
