# Peer review of "Health-Related Outcomes and Molecular Methods for the Characterization of A1 and A2 Cow’s Milk: Review and Update"

_vetsci, 2024, doi:10.3390/vetsci11040172_

Round 1

Reviewer 1 Report

Comments and Suggestions for Authors

The manuscript addresses a scientific topic of great relevance due to the implications it could have on public health. It is well written and structured, however, I make a series of suggestions to the authors that could improve it.

The first time the authors write the term in vitro (line 15) they do so with italics, however in the rest of the document they stop using this type of font. I recommend always using italics when writing terms in other languages.

The first time the authors write type 1 diabetes mellitus (line they define its abbreviation: (DM-1). It is understood that when abbreviations are defined it is because in the rest of the text they will continue to be used instead of the full term, however the authors write it in different ways throughout the text. I suggest homogenizing it as well as for the other cases in which abbreviations are indicated in the draft.

In contrast, the authors define some abbreviations (for example EFSA), for The European Food Safety Authority, although they do not mention them again in the rest of the document. I suggest avoiding it, as it is unnecessary.

In relation to the rest of the document, it seems appropriate in all its parts. However, being such a current topic and due to their relevance in relation to public health, the literature cited in the writing is, in my opinion, scarce. My arguments are:

·      106 documents are cited, covering a range of publication dates from 1984 to 2023. Of the total documents, 37 (34.9%) correspond to the last 5 years (without considering 2024, for which there are no citations), while that 52 documents (45%) are from the last decade.

·      The above demonstrates a good balance between the publication dates of the cited texts. However, when performing a search in Google Scholar using "casein a1" and "casein a2" as keywords, the result returned 376 results. This could suggest that the draft under analysis needs more citations, since it is a review article.

·      On the other hand, among those 376 documents that were obtained as a result of the Google Scholar search, 58 are reviews. This shows the importance of the topic and the need to delve deeper into what has been written about it.

·      Of the 376 search results, 13 correspond to the year 2024, 57 to publications from 2023 to date and 151 to publications from 2019 to date. Of these publications, 4, 14 and 36 are review articles for those same periods.

·      Finally, reference 27, which is obtained from the internet, is poorly referenced since the authors do not note its title (A2 Milk Facts), nor the author (Kimberly Yarris), nor the date of its publication (Feb 9, 2017). Furthermore, the cited document lacks scientific value since it is a document for dissemination and does not contribute anything extra to the writing.

Author Response

Answers to the reviewer 1

The manuscript addresses a scientific topic of great relevance due to the implications it could have on public health. It is well written and structured, however, I make a series of suggestions to the authors that could improve it.

The first time the authors write the term in vitro (line 15) they do so with italics, however in the rest of the document they stop using this type of font. I recommend always using italics when writing terms in other languages.

Answer: We have corrected this mistake. Thank you!

The first time the authors write type 1 diabetes mellitus (line they define its abbreviation: (DM-1). It is understood that when abbreviations are defined it is because in the rest of the text they will continue to be used instead of the full term, however the authors write it in different ways throughout the text. I suggest homogenizing it as well as for the other cases in which abbreviations are indicated in the draft.

Answer: We made all the correction.

In contrast, the authors define some abbreviations (for example EFSA), for The European Food Safety Authority, although they do not mention them again in the rest of the document. I suggest avoiding it, as it is unnecessary.

Answer: We deleted EFSA.

In relation to the rest of the document, it seems appropriate in all its parts. However, being such a current topic and due to their relevance in relation to public health, the literature cited in the writing is, in my opinion, scarce.

My arguments are:

  • 106 documents are cited, covering a range of publication dates from 1984 to 2023. Of the total documents, 37 (34.9%) correspond to the last 5 years (without considering 2024, for which there are no citations), while that 52 documents (45%) are from the last decade.
  • The above demonstrates a good balance between the publication dates of the cited texts. However, when performing a search in Google Scholar using "casein a1" and "casein a2" as keywords, the result returned 376 results. This could suggest that the draft under analysis needs more citations, since it is a review article.
  • On the other hand, among those 376 documents that were obtained as a result of the Google Scholar search, 58 are reviews. This shows the importance of the topic and the need to delve deeper into what has been written about it.
  • Of the 376 search results, 13 correspond to the year 2024, 57 to publications from 2023 to date and 151 to publications from 2019 to date. Of these publications, 4, 14 and 36 are review articles for those same periods.
  • Finally, reference 27, which is obtained from the internet, is poorly referenced since the authors do not note its title (A2 Milk Facts), nor the author (Kimberly Yarris), nor the date of its publication (Feb 9, 2017). Furthermore, the cited document lacks scientific value since it is a document for dissemination and does not contribute anything extra to the writing.

Answer: We replaced the reference 27. Also, we added supplementary references in our article.

Reviewer 2 Report

Comments and Suggestions for Authors

This is an important topic and careful review is essential, which is why I agreed to review the manuscript.

Detailed suggestions follow:

Throughout:  in vitro and in vivo should be italicized.

L21 and elsewhere:  For clarity, I think you should modify the sentence to say "...A2 milk may have potential health benefits beyond that of A1 milk..." since A1 milk also has health benefits (e.g. 13 essential nutrients).

L29:  is the word "new" needed?

L35:  following hypotheses that (not hypothesized)

L56:  food processing and digestion??

L63:  "poor" is a vague word.  Can you specify (e.g., conflicting, nonsignificant)?

L70:  instead of calving, say birth or "freshening" since you refer to all mammals.

L73:  since there is so much variability, please say "approximately" for components.

L77:  be careful--proteins have desirable behaviors as well.  Why not just say gelation, foaming, and other properties that may be considered desirable or undesirable in different applications?

L80:  remove "groups"

Perhaps note that there are genetic variants of all caseins and whey proteins.

L92:  and heat (perhaps mention that as well, since over-heating milk reduces cheese curd formation capabilities).

L93:  That sentence does not fit in that paragraph (and Ackerman is not the only one to have reported such information).  Perhaps remove that sentence and move the second sentence to the earlier paragraph about genetic variants. 

L98:  Cut the details about genetic variants (L81-95) out of the previous section and ONLY talk about it in the caseins section.  In other words, shorten the previous section and avoid redundancy in the casein section.

L110:  Bos taurus should be italicized.

L123:  Be careful:  A2 beta-casein in other species is not EXACTLY the same as cow A2.

L133:  You and I know that association does not mean causality, so it may be important to, somewhere, mention that.  Also plenty of peer-reviewed research has reported that dairy products are INVERSELY associated with a number of chronic diseases.  That may be worth mentioning as well.  

L141:  here you refer to A1/A1 and A2/A2 where previously you only used A1 or A2.  Perhaps explain earlier in the paper.

L151: Cite the references for impact of peptides on blood pressure, etc.  You need to explain WHY there would be any "intestinal discomfort" from consuming A1 milk.  Do not assume that the reader can know that.  Isn't it also true that, like with lactose maldigestion, there is a RANGE of discomfort (none - mild - moderate - severe) that can result in those who are sensitive (not everyone has issues with A1 milk).  What does BCM-7 actually do?  How?  Tell us more.

  L154:  what do you mean by bio-available?  Nutrients in A1 milk are also bio-available.  Clarify and cite.

L155:  Careful with wording:  it is NOT NECESSARY for MOST people to avoid A1.  And switching to goat, sheep, buffalo, etc. milk is not feasible for people on a tight budget or in food deserts.  

L171:  Now, here you show A1/A2 for the first time.  Again, this all needs to be explained earlier in the manuscript.

L172:  two phases?  I don't know what that means.  Clarify.  In fact, this whole paragraph seems a bit irrelevant to the rest of the section.  Re-work it to make it clearer.  Also clarify how it is relevant nutritionally, since that is the primary focus of your manuscript.

L187:  here in this paragraph is where you likely should mention association does not mean causality and highlight some of the peer-reviewed literature about milk's inverse association with CHD and DM-type 1 (likely predominantly A1 milk was involved in those studies--not always specified).  

L193:  much of that paragraph is redundant with earlier statements.  The last sentence in that paragraph probably belongs earlier, where I mentioned that you need to explain BCM-7 in more detail.

L200:  there is redundancy here as well.  MINIMIZE REDUNDANCY.

L213:  There is not enough compelling evidence presented here to say that.  Clarify this section if you want to convince us.

L243:  finish telling us what was concluded by Sun et al. before going into the autism papers (and separate paragraphs).

PERHAPS L253-266 belong earlier in the paper, where you explain BCM-7 more?

L272:  here you bring in allergenicity for the first time.  Maybe cow's milk allergy needs to be defined earlier in the paper then clarify if A1 and A2 differentiation is about allergenicity or digestion or or DM-1 or CHD or all of these.  Be careful not to confuse the reader...

L275:  now you bring in antioxidative functionality for the first time, without defining or fully explaining its implications.  How does that fit in?  

L277:  Now you bring in k-casein.  the section could be synthesized a little better (re-work it).

L295+:  Does this stuff belong earlier in the manuscript, where you will add more details to explain BCM-7?  Consider re-organization of the manuscript.

L316+:  Keep that section focused on the characterization methods.  Some of these paragraphs seem better suited for earlier in the manuscript.  Consider re-organization (and avoid redundancy).

L322:  Are there actually "health benefits" or is it really that the product may be better tolerated by people sensitive to BCM-7?  Please clarify the truth rather than the hype.  Don't the 13 essential nutrients in ALL milk that are beneficial to people (and only a fraction of the population have diagnosed intolerances to milk).

L325:  again, what are the "additional health benefits" beyond that of regular milk?

L327:  PERCEIVED health risks- not necessarily documented in case-controlled studies (controversial).

L334:  finally, here you get to what I have been pushing you to say...  Some of that should be earlier in the manuscript, or at least use more of the words "purported", "supposed", "claimed" to refer to not necessarily substantiated claims by A2 companies and researchers.

L338+: this seems very out of place.  It was reading like you were coming to the end, but this seems like stuff that belongs in the beginning.  Re-organize.

L359:  new term (CSN2) needs definition.

L405:  tell the reader why this would be necessary (validation of claims of A2/A2 milk?).

The conclusion seems like it was written by a different person.  It needs revision after the manuscript is re-organized/revised.

Author Response

Answers to the reviewer 2

This is an important topic and careful review is essential, which is why I agreed to review the manuscript.

Detailed suggestions follow:

Throughout:  in vitro and in vivo should be italicized.

Answer: We corrected!

L21 and elsewhere:  For clarity, I think you should modify the sentence to say "...A2 milk may have potential health benefits beyond that of A1 milk..." since A1 milk also has health benefits (e.g. 13 essential nutrients).

Answer: We included this in our sentence.

L29:  is the word "new" needed?

Answer: We rephrase this sentence.

L35:  following hypotheses that (not hypothesized)

Answer: We corrected. Thank you very much.

L56:  food processing and digestion??

Answer: We included this in out sentence.

L63:  "poor" is a vague word.  Can you specify (e.g., conflicting, nonsignificant)?

Answer: We change poor with conflicting.

L70:  instead of calving, say birth or "freshening" since you refer to all mammals.

Answer: We clarify this sentence.

L73:  since there is so much variability, please say "approximately" for components.

Answer: We included this word in our sentence.

L77:  be careful--proteins have desirable behaviors as well.  Why not just say gelation, foaming, and other properties that may be considered desirable or undesirable in different applications?

Answer: We rephrase this sentence.

L80:  remove "groups"

Answer: This word was removed.

L92:  and heat (perhaps mention that as well, since over-heating milk reduces cheese curd formation capabilities).

Answer: We rephrase this sentence.

L93:  That sentence does not fit in that paragraph (and Ackerman is not the only one to have reported such information).  Perhaps remove that sentence and move the second sentence to the earlier paragraph about genetic variants. 

Answer: We move this sentence to the genetic variants section.

L98:  Cut the details about genetic variants (L81-95) out of the previous section and ONLY talk about it in the caseins section.  In other words, shorten the previous section and avoid redundancy in the casein section.

Answer: We remove this senteces.

L110:  Bos taurus should be italicized.

Answer: We corrected this.

L123:  Be careful:  A2 beta-casein in other species is not EXACTLY the same as cow A2.

Answer: We remove the other species from the sentence. Thank you for the correction.

L133:  You and I know that association does not mean causality, so it may be important to, somewhere, mention that.  Also plenty of peer-reviewed research has reported that dairy products are INVERSELY associated with a number of chronic diseases. That may be worth mentioning as well.  

Answer: We clarified this sentence.

L141:  here you refer to A1/A1 and A2/A2 where previously you only used A1 or A2.  Perhaps explain earlier in the paper.

Answer: We clarified this earlier in the document.

L151: Cite the references for impact of peptides on blood pressure, etc.  You need to explain WHY there would be any "intestinal discomfort" from consuming A1 milk.  Do not assume that the reader can know that.  Isn't it also true that, like with lactose maldigestion, there is a RANGE of discomfort (none - mild - moderate - severe) that can result in those who are sensitive (not everyone has issues with A1 milk).  What does BCM-7 actually do?  How?  Tell us more.

Answer: We remove that sentence because of lack of relevant citation sources.

"In vitro studies have demonstrated that BCM-7 stimulates the secretion of gastrointestinal mucin [82-84]. Oral administration of BCM-7 [85] and consumption of A1 β-CN [86] induced inflammatory responses in mice. The A1 milk causes gastrointestinal inflammation, discomfort in the digestive tract, and/or delayed gastrointestinal transit in humans [87-89] and rats [90]"

  L154:  what do you mean by bio-available?  Nutrients in A1 milk are also bio-available.  Clarify and cite.

Answer: We remeoved that sentence because you have right. Both type of milk are bio-available.

L155:  Careful with wording:  it is NOT NECESSARY for MOST people to avoid A1.  And switching to goat, sheep, buffalo, etc. milk is not feasible for people on a tight budget or in food deserts.  

Answer: "You have a right," is an improper sentence. Therefore, we have removed it from our article.

L171:  Now, here you show A1/A2 for the first time.  Again, this all needs to be explained earlier in the manuscript.

Answer: We have explained along with a figure in our article.

L172:  two phases?  I don't know what that means.  Clarify.  In fact, this whole paragraph seems a bit irrelevant to the rest of the section.  Re-work it to make it clearer.  Also clarify how it is relevant nutritionally, since that is the primary focus of your manuscript.

Answer: We rephrase this sentence.

L187:  here in this paragraph is where you likely should mention association does not mean causality and highlight some of the peer-reviewed literature about milk's inverse association with CHD and DM-type 1 (likely predominantly A1 milk was involved in those studies--not always specified).  

Answer: We included in our article your suggestion. It's important to note, however, that an association between the peptide and these conditions does not necessarily indicate causality. Additionally, there is some peer-reviewed literature that suggests an inverse relationship between milk consumption and the risk of CHD and DM-type 1.

L193:  much of that paragraph is redundant with earlier statements.  The last sentence in that paragraph probably belongs earlier, where I mentioned that you need to explain BCM-7 in more detail.

Answer: We have removed that paragraph to avoid redundancy.

L200:  there is redundancy here as well.  MINIMIZE REDUNDANCY.

Answer: We have removed that paragraph to avoid redundancy.

L213:  There is not enough compelling evidence presented here to say that.  Clarify this section if you want to convince us.

Answer: We have revised every sentence in this paragraph to make them more persuasiv.

L243:  finish telling us what was concluded by Sun et al. before going into the autism papers (and separate paragraphs).

Answer: We have covered all aspects of Sun et al.'s conclusion in our article.PERHAPS

L253-266 belong earlier in the paper, where you explain BCM-7 more?

Answer: We have relocated this paragraph to an earlier section in the paper.

L272:  here you bring in allergenicity for the first time.  Maybe cow's milk allergy needs to be defined earlier in the paper then clarify if A1 and A2 differentiation is about allergenicity or digestion or or DM-1 or CHD or all of these.  Be careful not to confuse the reader...

Answer: The allergenicity of milk was covered earlier in our paper.

L275:  now you bring in antioxidative functionality for the first time, without defining or fully explaining its implications.  How does that fit in?  

Answer: We included in our paper the implications of the antioxidant properties of milk.

L277:  Now you bring in k-casein.  the section could be synthesized a little better (re-work it).

Answer: We modified this section.

L295+:  Does this stuff belong earlier in the manuscript, where you will add more details to explain BCM-7?  Consider re-organization of the manuscript.

Answer: We reorganized the manuscript.

L316+:  Keep that section focused on the characterization methods.  Some of these paragraphs seem better suited for earlier in the manuscript.  Consider re-organization (and avoid redundancy).

Answer: We reorganized the manuscript and we avoided the redundancy.

L322:  Are there actually "health benefits" or is it really that the product may be better tolerated by people sensitive to BCM-7?  Please clarify the truth rather than the hype.  Don't the 13 essential nutrients in ALL milk that are beneficial to people (and only a fraction of the population have diagnosed intolerances to milk).

Answer: We deleted this sentence.

L325:  again, what are the "additional health benefits" beyond that of regular milk?

Answer: We deleted this expression.

L327:  PERCEIVED health risks- not necessarily documented in case-controlled studies (controversial).

Answer: It was a mistake on our part to use such a harsh expression considering the general trend of the data presented. Thank you for your attention regarding this matter

L334:  finally, here you get to what I have been pushing you to say...  Some of that should be earlier in the manuscript, or at least use more of the words "purported", "supposed", "claimed" to refer to not necessarily substantiated claims by A2 companies and researchers.

Answer: We reorganized the manuscript. This sentence earlier in the manuscript.

L338+: this seems very out of place.  It was reading like you were coming to the end, but this seems like stuff that belongs in the beginning.  Re-organize.

Answer: We reorganized the manuscript. This sentence earlier in the manuscript.

L359:  new term (CSN2) needs definition.

Answer: β-CN is encoded by the gene CSN2.

"The four types of casein (αs1-, αs2-, β-, and κ-CN) are encoded by the genes CSN1S1, CSN1S2, CSN2, and CSN3, respectively"

L405:  tell the reader why this would be necessary (validation of claims of A2/A2 milk?).

Answer: We explained why it's important to identify the β-CN genetic variations in skim milk powder.

Round 2

Reviewer 2 Report

Comments and Suggestions for Authors

The manuscript is much improved, but I have some additional recommendations:

L55:  select a different word than pivotal.

Close to L99, explain A1/A1, A1/A2 and A2/A2 genetics and A1 and A2 designations.

Did you get permission from [33] and [40, 41] to use their figures?

L183:  so what?  what are the implications?  

L197:  perhaps comment on the low number of subjects in study.

L213:  elaborate on implications to human health.

L238:  exponential seems high.  Explain or change the word.

L251:  include some of those references in citation.

Lines 340-348 and 349-367 do not seem necessary (they do not fit perfectly under the section heading).  Consider shortening a bit more or complete removal.

Comments on the Quality of English Language

Improved

Author Response

Answers to the reviewer 2

L55:  select a different word than pivotal.

Answer: We change pivotal with vital.

Close to L99, explain A1/A1, A1/A2 and A2/A2 genetics and A1 and A2 designations.

Answer: We included, close to L99, the following sentences:"Cows can express both types of genes in their milk, either homozygous or heterozygous with allelic co-dominance. This means that a cow with the homozygous  A1A1 genotype produces pure A1 milk, while A2A2 genotype produces pure A2 milk. However, if an individual is heterozygous for this gene, A1A2, then the milk produced will contain a mix of the two forms β-CN"

Did you get permission from [33] and [40, 41] to use their figures?

Answer: We have requested requested permission to use and cite the source for the figures, but we still haven’t receive a response. However, as we understood, some journal's policy does not require permission for this purpose (MDPI | Open Access Information).

L183:  so what?  what are the implications?  

Answer: We rephrase this sentence for more explanations.

L197:  perhaps comment on the low number of subjects in study.

Answer: We included this comment in our sentence.

L213:  elaborate on implications to human health.

Answer: We included in our sentece an explanation regarding the impact on human digestion and overall well-being.

L238:  exponential seems high.  Explain or change the word.

Answer: We rephrased that sentence.

L251:  include some of those references in citation.

Answer: We included those references in our article.

Lines 340-348 and 349-367 do not seem necessary (they do not fit perfectly under the section heading).  Consider shortening a bit more or complete removal.

Answer: We deleted the lines 340-348 and 349-367
